# Façades-as-a-Service: The Role of Technology in the Circular Servitisation of the Building Envelope

Juan F. Azcarate-Aguerre [1,*], Tillmann Klein [1], Thaleia Konstantinou [1] and Martijn Veerman [2]

1 Chair Building Product Innovation, Department of Architectural Engineering + Technology, Faculty of Architecture and the Built Environment, Delft University of Technology, Julianalaan 134, 2628 BL Delft, The Netherlands; t.klein@tudelft.nl (T.K.); t.konstantinou@tudelft.nl (T.K.)
2 Alkondor Hengelo B.V., Wegtersweg 7-19, 7556 BP Hengelo, The Netherlands; m.veerman@alkondor.nl
* Correspondence: j.f.azcarateaguerre@tudelft.nl; Tel.: +31-61899-8904

**Featured Application: The research proposes a strategic path to align future development in façade and façade-integrated technologies with the new economic, legal, and organizational requirements of a more sustainable, circular, and performance-based façade industry.**

**Abstract:** The servitisation of the built environment, through the implementation of product–service systems, is considered a promising business strategy to achieve a circular economy transition. This servitisation faces a number of practical challenges, among them the technological readiness and effective integration and application of existing and emerging products, manufacturing processes, and digital monitoring and management tools. The research builds on targeted literature review, and on a research-through-design approach based on full-scale pilot projects developed in an ongoing feedback loop between researchers, planners, and industry partners representing both the demand and supply sides of the façade industry in the Netherlands. The paper analyses the technical implementation challenges currently preventing the façade industry from adopting performance-based contracts. It then proposes the roles that existing and emerging digital design and engineering technologies, manufacturing processes, and asset management systems can play in the development, implementation, and fulfilment of such contracts. The paper proposes a multi-stakeholder, systemic model for the development and application of façade technologies capable of overcoming many of the technical implementation barriers to the delivery of performance-based contracts for integrated facades. From this it concludes that an effective development of building technologies should strategically align with the solving of economic and contractual challenges such as circularity-readiness, profitability, risk distribution, legal demarcation, performance monitoring, and residual value stewardship. The resulting framework provides a strategic and conceptual basis for the development of circularity-enabling façade technologies, accounting for the diverse and sometimes conflicting interests of the multitude of stakeholders involved throughout a project's lifecycle. The framework aims to support planners, manufacturers, and builders accelerate the circular deep energy renovation of the built environment while also exploring new business opportunities.

**Keywords:** façade engineering; circular economy; product–service systems; energy renovation; built environment; performance contracts; facades-as-a-service; service integration

## 1. Introduction

The contribution of the built environment and the architecture, engineering and construction (AEC) sector to global environmental impact indicators is profound and well-documented [1–3]. In this context, the systemic transition theories of circular economy (CE) and product–service systems (PSS) have emerged as promising and potentially synergetic strategies to limit or reverse this environmental, economic, and societal impact [4–6]. Two fundamental aspects of CE theory—the dematerialization of economic activities and the

conservation of material resources within closed industrial loops—could have a considerable positive impact on the reduction of environmental degradation caused by activities related to the construction sector and the built environment.

Product–service systems (PSS) are a set of business and industrial strategies that propose to shift the key value in economic transactions away from the sale and transfer of material products and instead focus it on the effective and ongoing provision of performance services. Such strategies are often linked to the CE discussion due to the natural alignment of their incentive structures, value (co-)creation objectives, and potential dematerialization effects [7–10].

The implementation of CE and PSS principles requires, in practice, a broad restructuring of the economic, legal, financial, and technological foundations on which our current economic and industrial systems are based [11–13]. These challenges are particularly complex in the case of the construction sector, which is defined by large volumes of material use, long project timeframes spanning decades or even centuries, and specific legal and financial characteristics which are deeply embedded in our socio-cultural and economic environment, and which form one of the pillars of the global economy [14–16]. Along the transition towards a circular façade economy, the role of technical innovation is essential and diverse. Aspects such as the design and engineering of circular products and manufacturing processes, the monitoring of ongoing performance indicators, and the long-term tracking of embodied materials and maintenance schedules have a determinant effect on the technical feasibility, legal and managerial viability, and financial bankability of PSS offerings [17–19]. In order to pursue PSS ambitions, manufacturing companies must build significant capacity and take on extended value chain activities beyond their traditional front-office sales [20]. Furthermore, even if a PSS-based value chain is achieved, critical and continuous review of product manufacturing, service delivery, and reverse logistics processes is required in order to translate circular intent into actual resource circularity [21]. All these circularity-enabling products and processes can be translated into a new set of functional requirements for facades and other building products. Such functional requirements can conceptually determine the contribution of specific components and practices to achieving specific circularity and/or servitisation objectives. They can also be used as the primary procurement input on which designers, engineers, planners, manufacturers, and builders can develop specific and strategically aligned technical solutions.

This study is one of the outcomes of the five-year façade leasing project, in which various consortia of academic and industry partners worked on the development and execution of two full-scale façade servitisation pilot case-studies. During the process the research team explored the different dimensions of the implementation challenge, from the technical to the organisational, financial, and legal. The innovative contribution of this work is the integration between technical façade functionalities and applications [22,23], economic and organisational challenges of PSS [19,20], and strategic circular economy objectives [24,25]. By collecting these theoretical priorities into a single conceptualisation framework—supported by current façade industry examples—the full picture of a PSS-based CE transition can be more easily understood, mapped, and co-developed by both academy and industry. In the increasingly mainstream discussion on PSS and CE, a growing number of manufacturers are marketing solutions claiming to achieve material circularity. In the absence of final consensus on how to measure and monitor effective material circularity, a framework is needed to relate services, functionalities, and technologies on the path toward delivering circular facade services. This framework can be used by both technology suppliers and project commissioners to define the changing role of technology in the delivery of circular building envelope solutions.

## 2. Hypothesis

The hypothesis underlying this study is that—in the transition towards a circular and performance-based façade economy—a new set of functional façade requirements is needed. This hypothesis follows from transitional PSS pathways reported by other manufacturing

sectors with more extensive experience along the servitisation path [26–29]. The traditional, product-based technical solutions of a linear façade economy are commissioned and paid for by the building investor/owner and in the final interest of the building user or the facility manager. These façade functions focus on facilitating a safe, comfortable, aesthetically pleasing, and in some cases environmentally efficient indoor space for the building users, by deploying a range of physical and digital technical solutions [22,30–33]. Once the construction phase is finalized these technical components must be maintained by a facility management team, frequently with limited or no further involvement from the original component manufacturers or from other technical experts involved in the planning and construction phases [34–36].

A new set of circularity-enabling façade functions must, in contrast, reflect and enable the long-term interests of a multitude of stakeholders, beyond the strictly necessary user-focused façade functions of the linear economy. These functions must also focus on the long-term preservation of the value embodied by the building through technical updating and upgrading to meet ever-changing internal and external demands [37]. Examples of such extended functions include digital building information modelling (BIM) twins to allow live tracking of components and materials; tagging technologies to facilitate asset management by financiers and facility managers; remote monitoring technologies to enable proactive maintenance schedules, among many others [26,38]. These extended circular functions are no longer strictly in the sole interest of the building owner or building user. They must therefore be commissioned and financed in a collaborative model that guarantees long-term value co-creation and sharing of technical responsibilities and economic incentives, while safeguarding the material stewardship and carbon efficiency interests of society as a whole. In this path, the study proposes to modify and expand upon the existing framework of technical functional requirements in order to unlock and enhance material circularity potential through extended stakeholder incentives and liabilities.

## 3. Materials and Methods

The study is based on three levels of research, bridging the gap between theory and practice. Due to the explorative nature of this research and the lack of directly comparable sources and references, the study instead draws inspiration from research in other manufacturing fields and tests the circular premises of this research against practical façade pilot projects developed in collaboration with large consortia involving dozens of industry partners and professional experts. The first step is a (non-exhaustive) literature study and desk research, the second and third steps follow a research-through-design approach in which specific technical challenges to PSS and CE implementation are balanced against existing functional requirements and technical solutions in an iterative process involving both supply and demand stakeholders [39–42].

### 3.1. Literature Study

The study builds on existing CE and PSS literature to identify the servitisation process followed by manufacturing companies—in some cases within but mostly outside of the construction sector-on their path towards incremental servitisation. A significant body of knowledge has been gathered in the last couple of decades which identifies the key strategic choices, technological innovations, and organizational drivers and barriers, which have enabled companies to effectively adopt transitional or full PSS operational models [43–45]. The literature research focused on scientific articles and reports related to previous experience with circular economy, product–service systems (and servitisation), and stakeholders' dialogue in manufacturing industries, which were also the main keywords in the search concepts. Scientific databased, such as Scopus were used, as well as experience of national and international projects found in the respective databases, such as Cordis [46] and the EIT projects and publications library [47].

### 3.2. Stakeholder Mapping and Consultation

In order to identify the key aspects and requirements for the implementation of PSS models in the façade industry it is essential to collect the views and strategic priorities of the stakeholders' constellation. To this end, the study aimed, on the one hand, at mapping those stakeholders that are the potential adopters of a PSS—from both a supply and demand perspective. On the other hand, it aimed at consulting with them, in order to identify opportunities and bottlenecks in the process. Stakeholder mapping and consultation sessions were organized over a period of five years (between 2015 and 2020) through dozens of meetings with decision-makers, facility managers, end-users, designers and engineers, legal and financial advisors, manufacturers, builders, and system suppliers. General meetings were organised on a bi-monthly basis, with the participation of both demand and supply parties and addressing technical solutions to PSS implementation concerns identified during the planning phases of the pilot projects. Specific meetings with individual stakeholders-facility management, faculty end-users, central finance and legal departments, among others—were scheduled as required when facing discipline-specific challenges. These stakeholders were grouped into a sequence of consortia of academic and industry partners which participated in the development of the three stages of the "façade leasing" research project.

Drivers identified during this process include customer acquisition and retention through product decommoditisation, new revenue streams and financial stability, outsourcing of initial capital requirements and technical responsibility. Barriers to be overcome include the creation of legal contracts, financing and corporate structures, management practices, technological integration, and reverse logistics models for the recovery of materials and components. These organisational drivers and barriers have been described by Azcarate-Aguerre, et al. [11].

### 3.3. Case-Study Implementation through Full-Scale Field Prototypes

The case-study phase consisted of two field pilot prototypes [48] using fully operational TU Delft buildings as case studies during the façade leasing research project (2015–2020).

The project focused on (1) developing working, full-scale façade-as-a-service (FaaS) built prototypes, through which to identify technical drivers and barriers to implementation, and (2) finding organisational, managerial, and regulatory solutions to address these drivers and barriers, by exploring the legal, financial, and corporate implications of a PSS model for the contracting of facades-as-a-service (FaaS).

The two full-scale pilot prototypes built were:

*The EWI faculty building Façade Leasing technical mock-up (2017),* in which was explored the technological readiness of modular façade-integrated technologies to deliver energy savings and indoor-comfort improvements in a generic meeting and lecture room at the target building [49].

*The CiTG faculty building large-scale demonstrator prototype (2019),* in which were explored the broader systemic drivers and barriers to the implementation of façade-integrated technologies through a PSS contract, beyond the purely technical. The project resulted in the deep energy retrofit of over 2.600 m$^2$ of the target building's façade [50,51]. Figure 1 shows the new CiTG façade system, which includes centrally monitored and controlled operable windows, an upper ventilation window to enable night-cooling during warm summer nights, and vastly improved thermal performance of both framing and panelling.

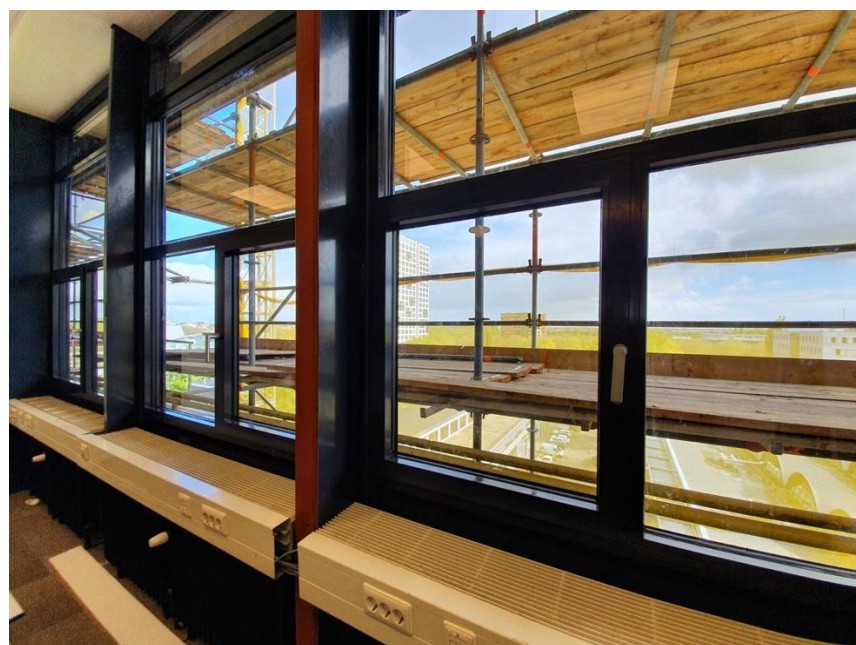

**Figure 1.** Interior photograph of the FaaS façade renovation at the CiTG faculty building large-scale demonstrator prototype, at TU Delft, in the Netherlands. Credits: Juan F. Azcarate-Aguerre, 2019.

In the process of engineering and developing the FaaS pilot projects we listened to the requests of diverse stakeholders for new and existing functional requirements which need to be physically or digitally in place in order to enable the construction, financing, operation, management and/or monitoring of a FaaS contract, as well as the high-value maintenance and recovery of components and materials. This research-through-design approach was based on an ongoing feedback loop between researchers and stakeholders during a design, engineering, and planning phase spanning almost a year—in the case of the small-scale EWI pilot project—and nearly two years in the case of the CiTG large-scale pilot project. The research-through-design feedback loop is illustrated in Figure 2.

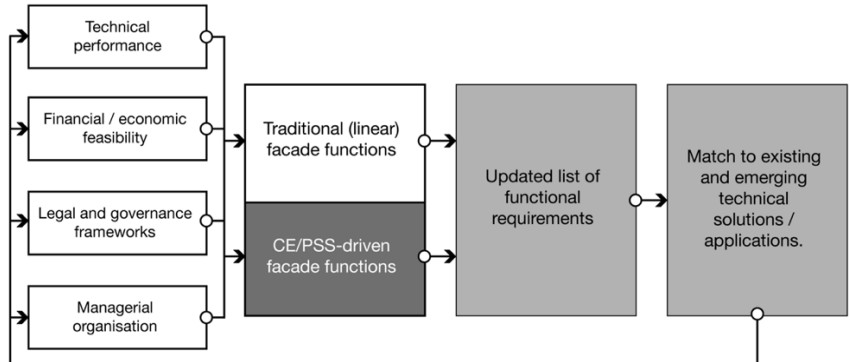

**Figure 2.** Research-through-design feedback loop in the multi-stakeholder co-creation process of the two façade leasing (FaaS) prototypes.

The study was carried out in an iterative feedback process bridging supply and demand, academy and industry. Due to the innovative and explorative nature of the Facades-as-a-Service model few references could be found beforehand to determine the exact technical and organisational challenges that would be faced in practice, or the entire range of technical solutions which would be needed to address these challenges. The research consortia, therefore, approached the problem as a practical engineering and business modelling process, testing the system through real-life case-study projects and implementing diverse façade-related technologies as specific and unexpected challenges

related to the FaaS model implementation and operation emerged. These iterative technical development steps, and their relevance to a circular façade economy, are reviewed and structured in a FaaS development matrix.

## 4. Results

Results are structured in accordance with the outcome of each methodological step:

### 4.1. Literature Summary

Based on literature references, two development paths are recognized as potentially leading to enhanced material circularity (CE-path) and servitisation of product manufacturers (PSS-path). These two complimentary paths constitute the strategic innovation direction for technological development in order to enable the CE and PSS transitions in the façade industry through specific and targeted steps:

*CE path (the Y-axis):* The key reference to establishing a clear theoretical framework for circular technological development are obtained from [24,25]. These sources propose three strategic paths: (A) smarter product use and manufacturing, (B) extend lifespan of product and its parts, and (C) useful (end-of-service) application of materials. The source also allocates ten "R" strategies to deal with materials and products and deliver the strategic goals (A to C mentioned above), Table 1.

**Table 1.** Circular strategies within the production chain, in order of priority. Adapted and summarised from Potting et al. [24].

| Building Phase | Produce | | | Construct | | | Use | | End-of-Service | |
|---|---|---|---|---|---|---|---|---|---|---|
| Circular strategy | Smart use and Manufacture | | | Extended lifespan | | | | | End-of-Service application | |
| "R" Strategy | R0. Refuse | R1. Rethink | R2. Reduce | R3. Reuse | R4. Repair | R5. Refurbish | R6. Remanufacture | R7. Repurpose | R8. Recycle | R9. Recover |

*PSS path (the x-axis):* Theory on the path followed by product manufacturers in their transitions towards service providers is obtained from Baines and Lightfoot [20,52]. In these sources the authors recognise and organise the incremental approach through which linear product manufacturers-whose role traditionally ends at the front-sales office-can gradually shift towards more integral performance-based contracts. Such contracts will enable them to retain responsibility (and in some cases also legal and economic ownership) over their products, while deriving long-term value from the service component and from the performance results delivered by these physical and digital products to their end customers. The steps along this process are identified as: (1) basic supply services such as the delivery of products and spare parts; (2) intermediate services involving the repair, overhaul, monitoring, and maintenance of the products; and (3) advanced services such as rental agreements, risk and revenue sharing, and revenue through use, Table 2.

### 4.2. Defining a Technological Development Matrix for New Stakeholder Circular Services in a Façade-as-a-Service System

The "FaaS technological matrix" (Figure 3) is the result of the intersection between performance contracting objectives and material circularity objectives. Traditional (linear) façade functions [22], can be located in the top-left corner of the matrix, as they: (A) deal mostly with the production phase in terms of circular economy strategy, offering limited opportunities for high value recovery or reprocessing; and (B) deal mostly with the product-

system end of the PSS spectrum. In other words, the delivery of products and replacement of certain components, but generally without extended customer service, system upgrading, performance delivery, or value co-creation intentions.

**Table 2.** Incremental servitisation process. Adapted from Baines & Lightfoot [20].

| Servitisation Level | Basic Services | Intermediate Services | | | | | Advanced Services | | | | |
|---|---|---|---|---|---|---|---|---|---|---|---|
| **PSS phase** | Supply | Repair | | Maintain | | | Manage | | | Operate | |
| **Tasks** | Supply product / Supply spare parts | Helpdesk / Repair | | Overhaul / Scheduled maintenance | | | Condition monitoring / Field services / Customer support agreement | | | Rental agreement / Risk and revenue sharing / Revenue through use | |

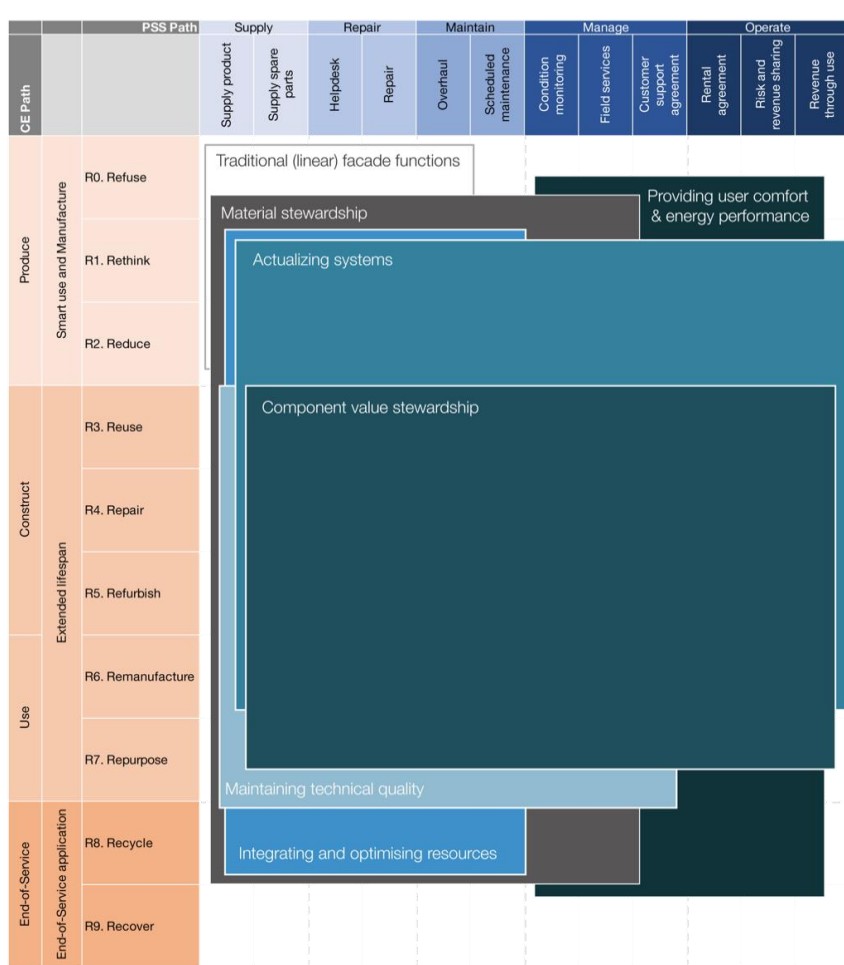

**Figure 3.** General fields of circular services enabled by a facades-as-a-service model.

As a first step, the matrix is used to organize the *circular services* demanded and delivered by the diverse consortium stakeholders, and therefore, provide a structure on which to map the more specific functional requirements and technical solutions. The purpose of this list is not to be exhaustive, as no present methodology would allow the creation of

an exhaustive list of all possible functional requirements nor technical solutions. Rather, the purpose of the list is to provide an overarching framework for the future development and/or application of building technologies. This framework is based not on gradual innovation and incremental improvement in a largely commoditized façade technologies market, but rather on the fulfilment of specific circularity- and sustainability-oriented service-delivery objectives as part of a collaborative and integral long-term value proposition.

### 4.3. Functional Requirements and Technical Solutions in the Transition to a Circular, Performance-Based Façade Industry. Lessons Learnt from the EWI and CiTG Pilot Projects

New functional requirements can be arranged along the PSS and CE paths on the matrix, resulting in a graphic impression of the scope of action enabled by each technology along both (interrelated) development paths. A breakdown of selected circular and performance-based functional requirements is shown in Figure 4, below. Examples of technical solutions addressing these technical requirements are shown to the right of the table.

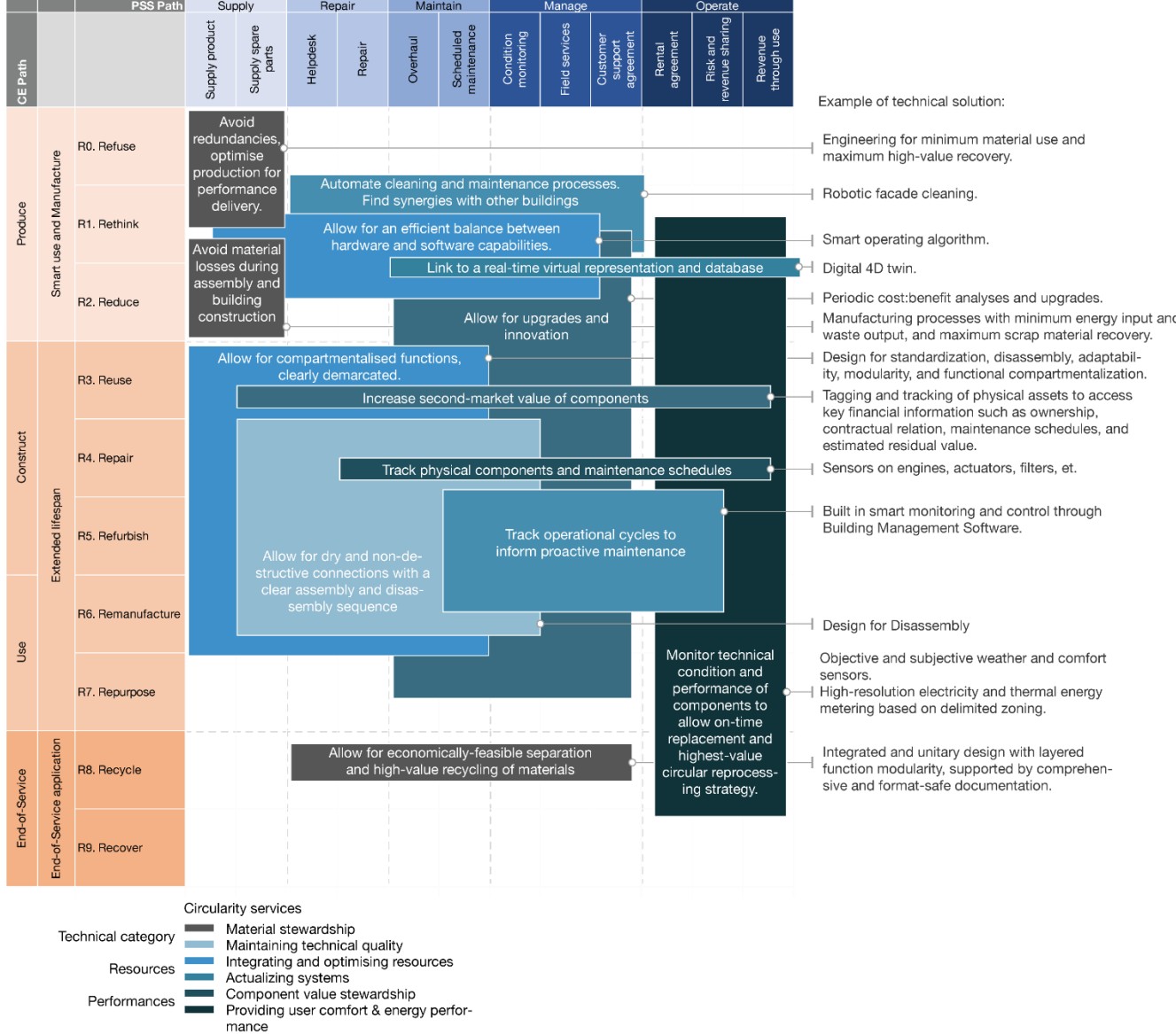

**Figure 4.** Selected new circularity-enabling façade functional requirements mapped according to the CE/PSS paths of the FaaS technological matrix. Examples of technical solutions to address these functional requirements are shown on the right.

The selected requirements and solutions illustrated above have been either implemented or considered for implementation in the two FaaS pilot projects. In some cases, the specific solution could not be implemented due to real-life constraints such as project budget, delivery timeframes, or full technological readiness. The technological readiness level (TRL) of most necessary façade-integrated technologies is high. In many cases, however, market-ready technologies are failing to reach full-scale market adoption due to a lack of demand or need for such technologies in a traditional procurement process.

The conceptualization of circular and performance-based functional requirements, not only during the early product development and project planning phases but throughout the building's service life, aims to solve the increasing issue of green-washing and circularity-washing in the construction sector. Within such a framework component manufacturers and suppliers should not limit their arguably circular offerings to products designed for disassembly or manufactured out of bio-based materials, but rather need to show proof that their product/service offering enables and aligns with the advanced functional requirements of the PSS and CE transitions throughout their operational lives.

## 5. Discussion

The innovative approach of this study is to propose a strategic path for the development of technical solutions to a new set of functional requirements as established by the economic and managerial priorities and demands of the diverse stakeholders involved in a potentially circular FaaS model contract. If diverse stakeholders are responsible for guaranteeing the ongoing performance of a façade over decades, then a new set of functional requirements is needed in order to reduce uncertainty and risk while monitoring actual delivered performance.

The outcome of this research is a strategic approach to the development of façade–integrated technologies, engineering and manufacturing processes, and asset management systems. These new practices must align with financially feasible material circularity and energy efficiency goals. Currently, the building industry in general and the façade industry specifically are characterised by a large number of small to medium enterprises working on different levels of gradual technological innovation and improvement. In terms of energy performance, the incremental improvement provided by better insulation or slightly more efficient heating and cooling systems has been increasingly encountering the economic law of diminishing returns [53–55]. Such diminishing returns become even more apparent when one considers the increasing value of larger volumes-or diversity-of material resources embedded in products, or the use of rare earth metals and other critical materials which are subject to limited global supply while crucial in the manufacturing of clean energy and smart building technologies [56–62]. Global supply-chain pressures, rising commodities markets, and volatile energy prices exacerbated by the COVID-19 pandemic have highlighted the fragility of material and component sourcing networks across all industries [63,64]. This is in turn leading to broader conscience among product manufacturers regarding the long-term financial and strategic cost of neglecting recovery of their material resources and remanufacturable cores [65–67].

A technological development path based on a transition towards façade servitisation represents a radical rethink of the economic incentive and decision-making processes which currently dominate the façade and construction industries. A shared long-term view of physical and digital building technologies aimed at maximising performance delivery provides the façade industry with a shared goal (and challenge) beyond that of competing among highly commoditized products and services with marginal technical distinctions. The technological implementation fields resultant from this study should not be seen as an exhaustive list of all components needed for an effective facades-as-a-service model, but rather as a strategic model according to which new technologies can be organised to meet actual value-chain challenges through technologies which are justified from-and required by-a whole multi-life-cycle perspective. The research demonstrates the relevance of collaborative systemic development and value creation by involving all relevant stakeholders

from the earliest planning phases. This aligns with the process described by other sources, in which functional requirements and technological readiness provide a starting point and practical dimension to the iterative process of co-designing a PSS offering [17,41,68].

Lastly, the paper highlights the relevance of a new way of understanding building procurement. The current one-way supply-chain in which all components are installed in order to fulfil demand-side needs or regulatory safety and health requirements, limits economic interest and financial investment from other stakeholders. New façade functionalities, such as design for disassembly or tracking of embodied components and materials, are not (yet) required in the interest of the client or regulators, but in the interest of the service provider whose components will benefit from a higher and more predictable residual value. Investment in such technical solutions, which are likely to increase the initial cost of CE- and PSS-enabled facades, must therefore be borne by the service provider or by the system's financiers in a shared co-investment and co-benefiting model.

## 6. Conclusions

The objective of this paper has been to outline a technological development path for facades and façade-integrated systems, in line with the changing requirements of the circular economy and product–service systems transitions. In other words, a system for understanding and organizing those new façade functionalities which might enable the feasibility and effectiveness of a FaaS contract.

Finding that existing technologies and emerging applications are enablers of PSS and a CE, our conclusions are that suppliers of building technologies aiming to engage in servitisation of their activities must extend their front-office operations from the sale of products to the ongoing delivery of measurable performance indicators. This transition significantly-though by no means solely-relies on technological innovation and integration.

Technical innovation targeting incremental improvement of component performance has so far often failed to reach mainstream implementation due to economic, social, political, or managerial barriers. The research-by-design process followed by this study and paper has exposed economic and contractual challenges such as profitability, risk distribution, legal demarcation, and performance monitoring, exacerbated by the long timescale of building projects, as key to the implementation of PSS models. Such models are, in turn, likely to safeguard CE objectives throughout the building's life cycle.

The systemic change introduced by the adoption of the technologies presented in this paper aims to align technological progress with a strategic view of CE and PSS goals. Such a perspective is expected to increase the chances of energy-efficient technologies achieving a wider market impact without resulting in a further increase in resource consumption and environmental degradation.

**Author Contributions:** Conceptualization, J.F.A.-A. and T.K. (Tillmann Klein); investigation and industry coordination, M.V.; methodology, T.K. (Thaleia Konstantinou); writing—original draft, J.F.A.-A.; writing—review and editing, T.K. (Tillmann Klein) and T.K. (Thaleia Konstantinou). All authors have read and agreed to the published version of the manuscript.

**Funding:** This research was funded by the European Institute of Innovation and Technology (EIT Climate-KIC): 190177. Full-scale prototypes were (co-)funded by Delft University of Technology, in collaboration with industry consortia. The APC was funded by Delft University of Technology.

**Institutional Review Board Statement:** Not applicable.

**Informed Consent Statement:** Not applicable.

**Acknowledgments:** The authors wish to thank EIT Climate-KIC, AluEco/VMRG, and Alkondor Hengelo B.V., as well as our partner consortium in both academy and industry, for their ongoing professional and financial support. This cross-disciplinary research would not be possible without the active and enthusiastic involvement of a large number of experts and organisations.

**Conflicts of Interest:** The authors declare no conflict of interest. The funders had no role in the design of the study; in the collection, analyses, or interpretation of data; in the writing of the manuscript, or in the decision to publish the results.

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
