# Peer review of "Facades-as-a-Service: The Role of Technology in the Circular Servitisation of the Building Envelope"

_applsci, doi:10.3390/app12031267_

Round 1
Reviewer 1 Report
The article is interesting and the content has good scientific merit. The structure of the paper is clear and straightforward. In addition, a good review of the relevant literature has been carried out. However, it is suggested to revise the text by simplifying or reducing, where possible, some sentences. The research results are clear, but their exposition could be improved with a summary table or a diagram.
Author Response
The text has been (briefly) revised for clarity. Figure 4 is intended to be the summary diagram.
Reviewer 2 Report
Review – Applied Sciences
Abstract
The methods used to conduct the study should be include. The key findings should be explicitly stated
Introduction
Provide references for this statement (lines 63-66) - Aspects such as the design and engineering of circular products and manufacturing processes, the monitoring of ongoing performance indicators, and the long-term tracking of embodied materials and maintenance schedules have a determinant effect on the technical feasibility, legal and managerial viability, and financial bankability of PSS offerings
Previous studies on the subject where not highlighted. As it stands, this section gives a more general background to the study rather than highlighting the contribution of the research. This should be done. The study's contributions need to be clearly delineated after presenting previous studies or existing knowledge and clearly situating this current study within existing knowledge.
Provide references for statement in lines 91 – 94 - Once the construction phase is finalized these technical components must be maintained by a facility management team, frequently with limited or no further involvement from the original component manufacturers or from other technical experts involved in the planning and construction phases.
The hypothesis section also does not demonstrate the theoretical foundation for the hypothesis. Studies which have investigated the existing frameworks must be presented. The gap must then be expanded on with evidence.
Methods
Details of the literature search should be provided. What kind of review was it? Systematic? Bibliometric? Etc. What kind of keywords were used? What kind of concepts and search results did the keywords yield? How many number of hits? What were the temporal boundaries of the search? How were the results of the literature review synthesized with the results of the stakeholder mapping?
The research design should also be appropriately described with theoretical support. How were the stakeholders identified, contacted and met? Focus groups, Delphi, workshops, symposiums? Also how was the sessions arranged during the pandemic? How many of the sessions were held? How many stakeholders were involved? How criteria informed the decision to involve them?
What ethical issues were considered in the stduy?
Figure 1 is not relevant. Delete.
Figures 2 and 3 should be explained.
The research phase should also be clearly and consistently described. In one section, the authors state, “Stakeholder mapping and consultation sessions were organized over a period of five years (between 2015 and 2020) through dozens of meetings w”. In another section, “This Research-through-Design approach was based on an ongoing feedback loop between researchers and stakeholders during a design, engineering, and planning phase spanning almost a year…”.
Do not use only numbers to refer to the authors in the text when they are part of the sentence. Include the name and then number.
Discussion
Tis section should compare and contrast the dings with existing similar studies and findings/frameworks.
Author Response
For clarity please refer to the attached document.

Reviewer 3 Report
The paper reports an interesting study dealing with the circular servitisation of the façade industry through a functional requirements framework that represents a quite innovative approach in the specific field. Despite the introduction summarizes most of the key elements and key theories from narrowing disciplinary fields, some additional references and explanations on the adopted assumptions would help the reader in contextualizing the effort made by the authors to transfer many consolidated elements from other sectors into the specific research field. That said, the adopted methodology maintains the novelty of the stated approach and combines different tools to support the study overall scope and objectives.
Contents are clearly presented and there is a good balance in the paper structure. The main achievements are coherently presented and discussed against barriers and limitations. A more explicit description of the value proposition with reference to the different stakeholders involved (which may have contrasting interests) would certainly enrich the text and its appeal for a broader audience. However, these are minor remarks considering the overall quality of the paper, just suggestions to further improve it.
Author Response
The paper reports an interesting study dealing with the circular servitisation of the façade industry through a functional requirements framework that represents a quite innovative approach in the specific field. Despite the introduction summarizes most of the key elements and key theories from narrowing disciplinary fields, some additional references and explanations on the adopted assumptions would help the reader in contextualizing the effort made by the authors to transfer many consolidated elements from other sectors into the specific research field. That said, the adopted methodology maintains the novelty of the stated approach and combines different tools to support the study overall scope and objectives. Additional references and contextualisation have been provided in the introduction and methodology sections.
Contents are clearly presented and there is a good balance in the paper structure. The main achievements are coherently presented and discussed against barriers and limitations. A more explicit description of the value proposition with reference to the different stakeholders involved (which may have contrasting interests) would certainly enrich the text and its appeal for a broader audience. However, these are minor remarks considering the overall quality of the paper, just suggestions to further improve it. An analysis of the value proposition, drivers and barriers from the perspective of different stakeholders has been described in previous publications of the authors. In this case it has been purposefully avoided to maintain the clarity of the specific ideas presented in this paper.
Round 2
Reviewer 2 Report
The paper has been improved. The quality of the figures (3 and 4) should be improved.